# Cutoff Points of Waist Circumference for Predicting Incident Non-Alcoholic Fatty Liver Disease in Middle-Aged and Older Korean Adults

**DOI:** 10.3390/nu14142994

**Published:** 2022-07-21

**Authors:** Jun-Hyuk Lee, Soyoung Jeon, Hye Sun Lee, Yu-Jin Kwon

**Affiliations:** 1Department of Family Medicine, Nowon Eulji Medical Center, Eulji University School of Medicine, Seoul 01830, Korea; swpapa@eulji.ac.kr; 2Department of Medicine, Hanyang University Graduate School of Medicine, Seoul 04763, Korea; 3Biostatistics Collaboration Unit, Department of Research Affairs, Yonsei University College of Medicine, Seoul 03277, Korea; jsy0331@yuhs.ac; 4Department of Family Medicine, Yongin Severance Hospital, Yonsei University College of Medicine, Yongin 16995, Korea

**Keywords:** abdominal obesity, body mass index, non-alcoholic fatty liver disease, waist circumference

## Abstract

This study aimed to determine the optimal cutoff value of waist circumference (WC) for predicting incident NAFLD. In this community-based prospective cohort study, we analyzed data from 5400 participants without NAFLD at baseline aged 40–69 years. NAFLD was defined as a NAFLD-liver fat score >−0.640. A Cox proportional hazards regression model was used to estimate the hazard ratio (HR) and 95% confidence interval (CI) for an association between body composition and NAFLD incidence. The predictive power of each body composition indicator was assessed by Harrell’s concordance index for Cox models. During a mean follow-up period of 12 years, there were 2366 new-onset NAFLD events. Compared with men with WC < 81 cm, the adjusted HR (95% CI) for incident NAFLD in those with WC ≥ 81 cm was 2.44 (2.23–2.67). Compared with women with WC < 78.5 cm, the adjusted HR (95% CI) for incident NAFLD in those with WC ≥ 78.5 cm was 2.54 (2.25–2.87). WC was the most significant risk factor for predicting incident NAFLD among body composition indicators in middle-aged and older Korean adults. The optimal WC cutoff point for predicting incident NALFD was 81 cm in men and 78.5 cm in women, which might assist in the early detection and prevention of NAFLD.

## 1. Introduction

Non-alcoholic fatty liver disease (NAFLD) is a condition resulting from fat accumulation in the liver and is one of the most common chronic liver diseases worldwide [1]. The global prevalence of NAFLD was estimated to be approximately 25% [2]. The prevalence of NAFLD in Asia is also around 27.4%, such as in many Western countries [3]. In 2021, the Global Burden of Diseases study reported a rapidly increasing burden of NAFLD associated with mortality and disability-adjusted life-years [4]. The established metabolic risk factors of NAFLD were found to be obesity, diabetes mellitus, insulin resistance, and metabolic syndrome [5]. These risk factors can be closely related to an unhealthy lifestyle: a high-calorie diet and a sedentary lifestyle. Moreover, there is no approved pharmacological treatment for NAFLD. Among intensive lifestyle modifications, including weight reduction and adequate physical activity, an ultimate goal of losing weight has been regarded as the primary strategy for NAFLD management [6].

Many previous studies have evaluated the relationship between body composition indicators and NAFLD [7,8,9,10,11,12]. Body mass index (BMI) and waist circumference (WC), widely used indicators for identifying obesity or abdominal obesity, are well-known risk factors for NAFLD [8,12]. In addition, low muscle mass or muscle mass index emerged as a risk factor for NAFLD [10,11]. Most studies suggested that high fat mass could better predict NAFLD incidence [7,9]. However, there have been few studies showing which indicator would mostly be associated with NAFLD incidence. Therefore, we aimed to verify the most related body composition indicator for predicting NAFLD incidence. Abdominal obesity in Koreans was defined as WC ≥ 90 cm in men and WC ≥ 85 cm in women, respectively [13]. Although previous studies revealed significant associations between abdominal obesity and NAFLD [12,14], the optimal WC range for predicting incident NAFLD is still unclear. Moreover, an ethnic-specific WC cutoff point for predicting incident NAFLD is needed. Therefore, we also aimed to determine the most appropriate cutoff points of WC for predicting incident NAFLD in Korean middle-aged and older adults.

## 2. Materials and Methods

### 2.1. Study Population

The Korean Genome and Epidemiology Study (KoGES) is a longitudinal prospective cohort study conducted by the Korea Centers for Disease Control and Prevention [15]. A total of 10,030 adults aged between 40 and 69 years and living in urban (Ansan) and rural areas (Ansung) were recruited into the KoGES_Ansan_Ansung cohort and evaluated biennially. This began with the baseline survey conducted in 2001–2002 and ended with the eighth follow-up in 2017–2018. Figure 1 shows the flowchart of the study population selection. Out of a total of 10,030 participants in the baseline survey, we excluded those with history of hepatitis (*n* = 423) and men whose alcohol consumption ≥ 30 g/day or women whose alcohol consumption ≥ 20 g/day (*n* = 964). Participants with: insufficient data to calculate NAFLD-liver fat score (*n* = 276); NAFLD at baseline (*n* = 2222); missing data of body composition (*n* = 40); and no history of follow-up after the baseline survey (*n* = 705) were also excluded from the study. Finally, a total of 5400 participants without NAFLD were included in the study. Informed consent was obtained from all participants. This study was approved by the Institutional Review Board of Nowon Eulji Medical Center (approval number: 2022-01-016).

### 2.2. Assessment of Non-Alcoholic Fatty Liver Disease

NAFLD was defined using the NAFLD-liver fat score that was calculated as follows: NAFLD-liver fat score = −2.89 + 1.18 × metabolic syndrome (yes: 1, no: 0) + 0.45 × diabetes mellitus (yes: 2, no: 0) + 0.15 × insulin (µIU/mL) + 0.04 × aspartate aminotransferase (AST) (U/L)—0.94 × AST/alanine aminotransferase (ALT). A NAFLD-liver fat score >−0.640 was considered indicative of having NALFD [16].

### 2.3. Assessment of Body Composition

Body weight and height were measured to the nearest 0.1 kg and 0.1 cm, respectively. BMI was calculated as a participant’s body weight (kg) divided by height in meters squared (m^2^). WC (cm) was measured at midpoint between the iliac crest and lowest rib. The measurement was performed at the end of the normal expiration with an inelastic tape adjacent to the skin without compressing it. The detailed protocol of physical measurement is described on the KoGES website (https://www.kdca.go.kr/contents.es?mid=a40504100200, accessed on 14 January 2022). Using the bioelectrical impedance analysis (BIA) (body composition analyzer, models ZEUS 9.9, JAWON MEDICAL CO., LTD, Seoul, Korea), body fat (%) and total muscle mass (TMM, kg) were measured according to the manufacturer’s guideline. We used height squared-adjusted TMM (TMM/Ht2) and BMI-adjusted TMM (TMM/BMI) for the analysis.

### 2.4. Covariates

After at least 5 min rest in the sitting position, systolic blood pressure (SBP) and diastolic blood pressure (DBP) were measured. Mean blood pressure (MBP) was calculated as DBP + 1/3 × (SBP—DBP). Smoking status was categorized into 4 groups: never smoker, ex-smoker, intermittent smoker, or everyday smoker [17]. The amount of alcohol intake (g/day) was assessed using the following calculation: the frequency of alcohol consumption (glasses/time) × 10 (g/per glass of drink) × alcohol consumption (times/month)/30 (days/month). Alcohol consumption was defined as participant who drank alcohol less than 30 g per day in men or less than 20 g per day in women. Physical activity was assessed by using metabolic equivalent of task (MET)-hours per week (METs-h/wk). The degree of physical activity was categorized into 3 groups: low (< 7.5 MET-h/wk), moderate (7.5–30 MET-h/wk), and high (> 30 MET-h/wk) [18]. Blood samples from each participant were collected after at least 8 h fasting and analyzed using a Hitachi 700-110 Chemistry Analyzer (Hitachi Co., Tokyo, Japan). Fasting plasma glucose (FPG), serum insulin, total cholesterol (TC), AST, ALT, and C-reactive protein (CRP) were determined. Total calorie intake was derived from a semi-quantitative food frequency questionnaire.

### 2.5. Statistical Analysis

Data are presented as the mean ± standard deviation (SD) for continuous variables and number (percentage) for categorical variables. The characteristics of men and women were compared using the independent *t*-test for continuous variables and chi-square test for categorical variables. Univariate and multivariate Cox proportional hazards regression models with person-years as the time metric were used to estimate hazard ratios (HRs) and 95% confidence intervals (CIs) of associations between body composition indices with regard to the outcomes. In multivariate model 1, we adjusted for age and sex. In multivariate model 2, we additionally adjusted for smoking, alcohol consumption, physical activity, and total calorie. In multivariate model 3, we additionally adjusted for MBP, FPG, TC, high sensitivity CRP, and ALT. The predictive powers and discriminatory capabilities for predicting incident NAFLD of five different body composition indices were assessed by Harrell’s concordance index for Cox models. To determine the optimal WC cutoff point for predicting NAFLD incidence, we used the method of Contal and O’Quigley [19]. When the optimal cutoff point that maximized HR of incident NAFLD was applied, there were significant differences in incident NAFLD in all participants. The Kaplan–Meier curve for evaluating incident NAFLD according to the optimal cutoff point was shown, and a log-rank test was performed. All statistical analyses were performed using SAS 9.2 (SAS Institute, Cary, NC, USA). Two-sided *p*-values < 0.05 were considered statistically significant.

## 3. Results

Table 1 shows the baseline characteristics of the study population. Out of the 5400 participants, of whom 2241 (41.5%) were men, the mean value of age and WC was 51.7 ± 8.8 years and 80.8 ± 8.1 cm, respectively. The mean age ± SD was 52.1 ± 8.8 years in men and 51.3 ± 8.8 years in women. The mean values of WC and BMI were 81.2 ± 6.9 cm and 23.5 ± 2.6 kg/m^2^ in men and 79.1 ± 8.7 cm and 24.1 ± 3.0 kg/m^2^ in women, respectively. During the mean (min, max) 12.0 (1.4, 16.0) years of follow-up, there were 2356 incident NAFLD events. Appendix A shows the baseline characteristics of the participants according to NAFLD incidence. The participants with incident NAFLD had higher WC, body fat, BMI, TMM, MBP, total cholesterol, FPG, CRP, AST, and ALT.

In Table 2, we found a relationship between five body composition indices (WC, body fat, BMI, TMM/Ht^2^, and TMM/BMI) and incident NAFLD. Per one increment of WC, body fat, BMI, and TMM/Ht^2^, the risk of incident NAFLD was significantly increased after adjusting for age, sex (for total), smoking, alcohol, exercise, total calorie intake, MBP, FPG, total cholesterol, CRP, and ALT. Meanwhile, per an increment of TMM/BMI, the risk of incident NAFLD was significantly decreased after adjusting for the same confounders. There were similar trends in men and women.

Then, we identified the c-indexes of the body composition indices that were significantly related to incident NAFLD in all participants (Figure 2a, b). Out of the five body composition indices, the Harrell’s concordance index (95% CI) of WC was 0.68 (0.67–0.69), which was significantly higher than that of the other body composition indicators in all participants (WC vs. body fat, *p* <0.001; vs. TMM/Ht^2^, *p* <0.001; vs. TMM/BMI, *p* <0.001, and vs. BMI, *p* = 0.001). The Harrell’s concordance index (95% CI) of WC was 0.68 (0.66–0.70) and 0.69 (0.67–0.70) in men and women, respectively. These were significantly higher than other indices in men (WC vs. body fat, *p* <0.001 in men and women; vs. TMM/Ht^2^, *p* <0.001 in men and women; vs. TMM/BMI, *p* <0.001 in men and women, and vs. BMI, *p* <0.001 in men and *p* <0.001 in women). The values of the c-index and 95% CI are presented in Appendix A.

Figure 3 shows the Kaplan–Meier curve for evaluating incident NAFLD defined by WC cutoff points. The optimal cutoff points were determined at 80.2 cm in all participants, 81 cm in men, and 78.5 cm in men, respectively, which were the most significant cutoff points to differentiate between participants with NAFLD incidence. Kaplan–Meier curve showed that a higher risk of cumulative NAFLD incidence was in WC ≥ 80.2 cm than in WC < 80.2 cm in all participants (*p* <0.001). Kaplan–Meier curve showed that higher risk of cumulative NAFLD incidence was in WC ≥ 81 cm than in WC < 81 cm in men (*p* <0.001) and WC ≥ 78.5 cm than WC < 78.5 cm in women (*p* <0.001).

Table 3 shows the relationship between WC cutoff points and incident NAFLD, using HR and 95% CI values. Compared with participants with WC < 80.2 cm, the HR (95% CI) for incident NAFLD in those with WC ≥ 80.2 cm was 2.75 (2.53–2.99) in the unadjusted model, 2.76 (2.53–3.01) in model one, 2.76 (2.53–3.02) in model two, and 2.44 (2.23–2.67) in model three. In men, compared with participants with WC < 81 cm, the HRs (95% CI) for incident NAFLD in those with WC ≥ 81 cm was 2.88 (2.50–3.32) in the unadjusted model, 2.87 (2.50–3.31) in model one, 2.99 (2.58–3.46) in model two, and 2.65 (2.28–3.08) in model three. In women, compared with participants with WC < 78.5 cm, the HR (95% CI) for incident NAFLD in those with WC ≥ 78.5 cm was 2.93 (2.62–3.28) in the unadjusted model, 2.77 (2.47–3.12) in model one, 2.77 (2.45–3.13) in model two, and 2.54 (2.25–2.87) in model three.

## 4. Discussion

In this longitudinal prospective cohort study, we identified that WC was the most significant risk factor for incident NAFLD among the various body composition variables in middle-aged and older Korean adults. Furthermore, we found that the optimal WC cutoff points for predicting NAFLD were below 81 cm in men and below 78.5 cm in women. These results were approximately 10% below the cutoff point of the WC value to define abdominal obesity.

Previous studies have revealed an association between body composition and the risk for NAFLD [7,20]. A recent cross-sectional study has shown that fat mass indices were more strongly associated with fibrosis than were muscle mass indices [7]. Other cross-sectional analyses in the Rotterdam Study reported that both high fat mass and lower lean mass were associated with NAFLD risk. However, fat mass better predicted the prevalence of NAFLD than did lean mass [9]. The Multi-Ethnic Study of Atherosclerosis showed that both WC and BMI were significantly associated with NAFLD risk. Furthermore, WC had the best discrimination for NAFLD [20]. A meta-analysis including 20 studies found that individuals with abdominal obesity measured by WC had a higher risk of NAFLD than did those with general obesity measured by BMI [12]. Our findings are in line with previous studies. We found that Harrell’s concordance index of WC for predicting incident NAFLD was significantly higher than that of other body composition indices.

Several possible mechanisms might support our findings. Abdominal obesity measured by WC might lead to hepatic pathologic changes and become a major risk factor for NAFLD [12]. Van et al. [21] demonstrated that visceral fat was independently associated with hepatic inflammation and fibrosis regardless of insulin resistance. Abdominal obesity is a key component of metabolic syndrome, which has been shown to be strongly associated with various metabolic diseases, including NAFLD [22,23]. Visceral adipose tissue would promote infiltration of macrophages and secretion of pro-inflammatory cytokines (tumor necrosis factor-α, interleukin-1β, interleukin-6, monocyte chemotactic peptide-1, and resistin) [24] and decrease adipocytokines secretion such as adiponectin that reduces insulin resistance and inflammation [25]. Inflamed adipose tissue could contribute to an influx of fatty acids to the liver. Moreover, decreases in adiponectin secretion might lead to ectopic fat accumulation in the liver via increases in dysregulation of free fatty acid oxidation and de novo lipogenesis [23,26].

Men are at higher risk of the development of NAFLD than women [27]. We found that the optimal cutoff points for incident NAFLD were WC < 81 cm (men) and < 78.5 cm (women), 10.0% (men) and 7.6% (women) below previous recommended WC in ethnic-specific metabolic syndrome criteria [28], respectively, which suggests that an intensive lifestyle intervention should be applied earlier in Korean men than in Korean women. Previous studies, however, have demonstrated that regardless of the same BMI, the body fat amount is significantly influenced by sex and race [29,30,31]. Therefore, the WC threshold for abdominal obesity could not be uniformly applicable to both men and women and different races. A Korean study including a total of 456 healthy participants aged 20–88 years indicated that the optimal cutoff point of WC in screening for NAFLD was ≥ 90 cm in men and ≥ 85 cm in women [13]. This was based on the point that the odds ratio for the risk of ≥ 2 components for metabolic syndrome except for WC increased rapidly [13]. NAFLD can be considered a hepatic manifestation of metabolic syndrome. Furthermore, it might be considered a precursor of this metabolic syndrome through insulin resistance [32]. Therefore, the optimal cutoff points of WC for predicting NAFLD might be lower than those of metabolic syndrome. In addition, we used data of a longitudinal prospective cohort data with a mean follow-up of 12 years. Since we determined the optimal cutoff value of WC to predict incident NAFLD in participants without NAFLD at baseline, the optimal cutoff value of WC in this study could be lower than the conventional cutoff point for defining abdominal obesity.

While BMI-adjusted TMM showed a negative association with the incidence of NAFLD, height squared-adjusted TMM showed a positive association with the incidence of NAFLD. It is generally accepted that skeletal muscle has a protective effect on maintaining insulin sensitivity [33]. However, in previous studies on the association between low muscle mass and NAFLD, height-adjusted appendicular skeletal muscle mass showed a positive relationship with the NAFLD and a positive correlation with insulin resistance unless body weight and/or BMI were adjusted for as a confounding variable [34,35]. The body fat would have had a greater effect on NAFLD development than on muscles. Considering visceral adipose tissue is metabolically active, our findings showing that waist circumference was the best predictive body composition indicator for incident NAFLD could support this hypothesis. A follow-up study is needed to identify whether people with low height-adjusted skeletal muscle mass have a higher risk of NAFLD than those with normal skeletal muscle mass by matching people with the same fat mass and bone mass.

Our study has several limitations. First, although we used a validated NAFLD prediction model [16], histological diagnosis and liver imaging were not available in this cohort. In a large epidemiological cohort study, ultrasonographic examination and histological confirmation could not be available due to high cost and ethical considerations. Second, measurement of body fat and muscle mass was conducted using the BIA. Modalities such as computed tomography, magnetic resonance imaging, and dual-energy X-ray absorptiometry are considered more reliable tools for assessing body composition. However, BIA is a relatively simple, inexpensive, and non-invasive technique to measure body composition and is more suitable in larger epidemiological studies [36]. Furthermore, we could only obtain the data about total skeletal muscle mass from the KoGES. Third, parameters assessed for body composition were not updated during the follow-up period. Therefore, baseline exposure could not have reflected longitudinal dynamic changes in body compositing with aging. Fourth, we could not consider other effects such as lifestyle changes or therapeutic interventions. Finally, our results might not be generalized to other ethnic groups, and similar studies should be conducted on other ethnicities.

Despite these limitations, this study has several strengths. First, to the best of our knowledge, this is the first study to identify the optimal WC cutoff point for incident NAFLD using large prospective cohort data with a long follow-up period. Additionally, this is a large population-based study using well-examined data to ensure the statistical power and reliability of our results.

## 5. Conclusions

This large population-based cohort study found that WC was the most powerful and significant risk factor for predicting incident NAFLD in middle-aged and older Korean adults. Furthermore, WC < 81 cm in men and < 78.5 cm in women were found to be the optimal cutoff points for predicting NAFLD incidence. Managing WC could be an effective strategy for the early detection and prevention of NAFLD. More well-designed studies in other countries and ethnic groups with sufficient laboratory and liver-imaging findings are needed to identify the optimal WC cutoff points for preventing NAFLD. Additionally, more active lifestyle interventions promoting weight loss through a healthy diet and physical activity are needed for maintaining liver health.

## Figures and Tables

**Figure 1 nutrients-14-02994-f001:**
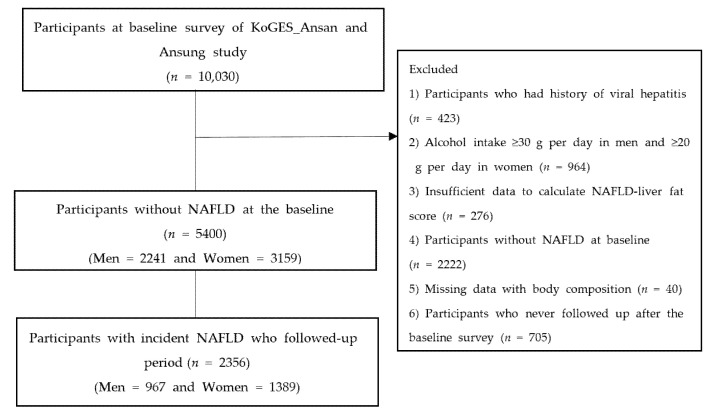
Flow chart of the study population.

**Figure 2 nutrients-14-02994-f002:**
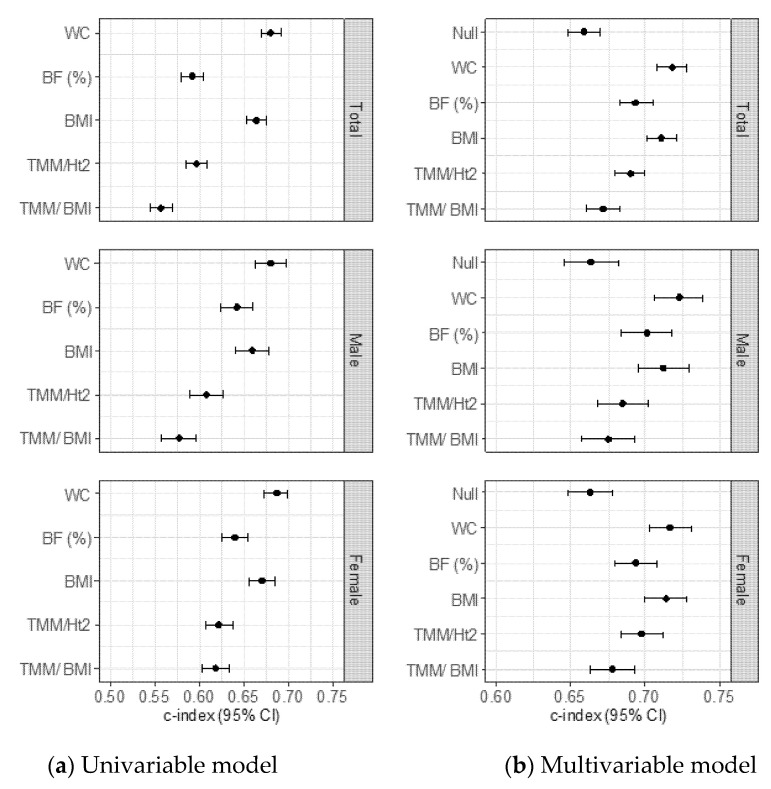
Harrell’s concordance index and 95% Confidence intervals for predicting NAFLD incidence of five different body composition indices. (**a**) Unadjusted model in total population, men and women, (**b**) Adjusted for age, sex (in total), smoking status, alcohol drinking status, physical activity, total calorie intake, mean blood pressure, fasting plasma glucose, CRP, and alanine aminotransferase in total participants, men and women. Abbreviation: BF, Body fat; BMI, Body mass index; CRP, C-reactive protein; TMM/Ht^2^; Total muscle mass divided by squared of height; TMM/BMI, Total muscle mass divided by body mass index

**Figure 3 nutrients-14-02994-f003:**
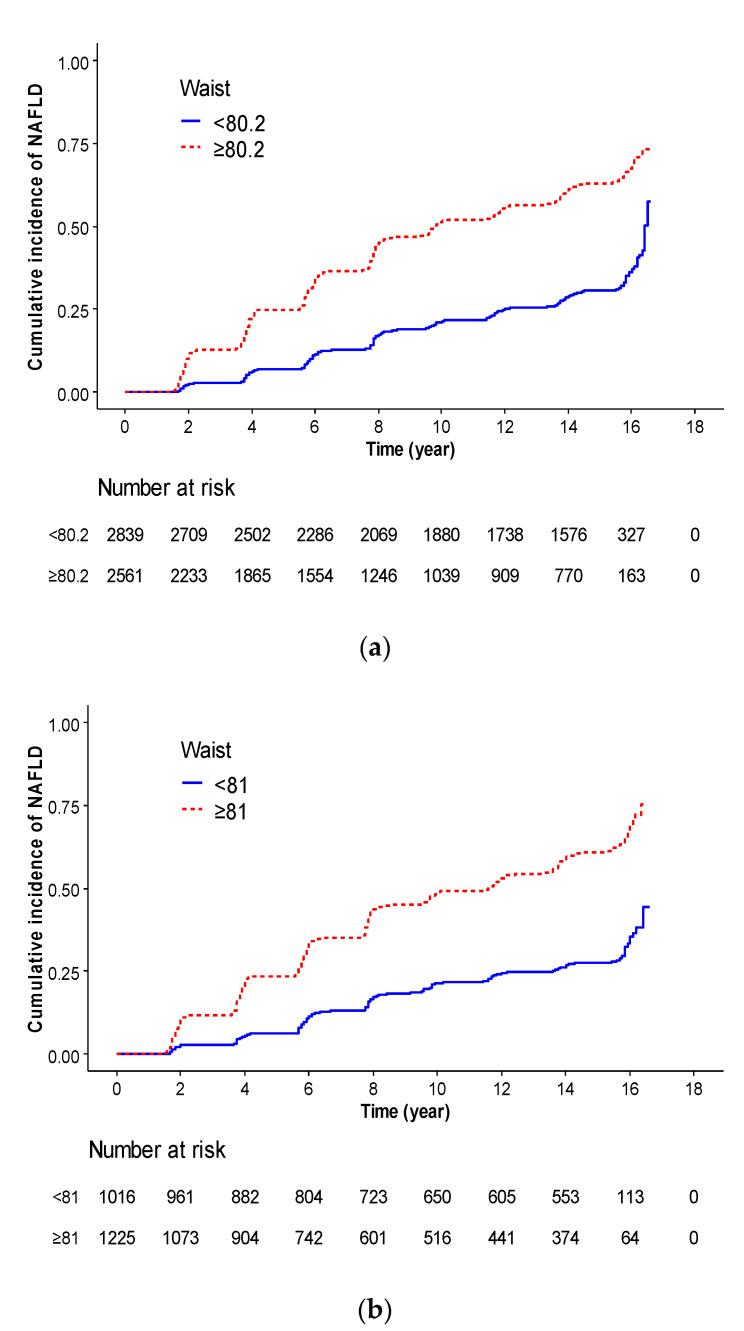
Kaplan–Meier curve for evaluating incident NAFLD identified by waist circumference cutoff points. (**a**) Total participants, < 80.2 cm (blue line) vs. ≥ 80.2 cm (red dot line), (**b**) Men, < 81 cm (blue line) vs. ≥ 81 cm (red dot line), (**c**) Women, < 78.5 cm (blue line) vs. ≥ 78.5 cm (red dot line)**.**

**Table 1 nutrients-14-02994-t001:** Baseline characteristic of the study population.

Variables	Total (*n* = 5400)	Men (*n* = 2241)	Women (*n* = 3159)	*p*
Age, years	51.7 ± 8.8	52.1 ± 8.8	51.3 ± 8.8	0.001
WC, cm	80.0 ± 8.1	81.2 ± 6.9	79.1 ± 8.7	<0.001
Body fat, %	26.3 ± 7.2	20.4 ± 4.9	30.5 ± 5.3	<0.001
BMI, kg/m^2^	23.9 ± 2.9	23.5 ± 2.6	24.1 ± 3.0	<0.001
TMM, kg	42.0 ± 7.6	48.8 ± 6.0	37.2 ± 4.2	<0.001
Height-adjusted TMM, kg/m^2^	16.5 ± 1.7	17.6 ± 1.5	15.7 ± 1.2	<0.001
BMI-adjusted TMM, m^2^	1.8 ± 0.3	2.1 ± 0.2	1.6 ± 0.2	<0.001
Body weight, kg	60.5 ± 9.1	65.1 ± 8.8	57.2 ± 7.8	<0.001
Smoking status, *n* (%)				<0.001
Non-smoker	3561 (66.8)	578 (25.9)	2983 (96.2)	
Ex-smoker	683 (12.8)	653 (29.3)	30 (1.0)	
Intermittent smoker	114 (2.1)	89 (4.0)	25 (0.8)	
Every day smoker	973 (18.3)	910 (40.8)	63 (2.0)	
Physical activity, *n* (%)				<0.001
Low (< 7.5 METS-h/wk)	392 (7.6)	121 (5.6)	271 (8.9)	
Moderate (7.5–30 METS-h/wk)	3148 (60.6)	1241 (57.8)	1907 (62.7)	
High (> 30 METS-h/wk)	1651 (31.8)	787 (36.6)	864 (28.4)	
Alcohol drinking, yes, *n* (%)	2305 (43.0)	1440 (64.7)	865 (27.7)	<0.001
Total energy intake, kcal/day	1937.1 ± 706.1	1997.7 ± 665.8	1894.1 ± 730.4	<0.001
MBP, mmHg	94.0 ± 12.6	95.6 ± 11.9	92.8 ± 13.0	<0.001
FPG, mg/dL	82.6 ± 11.8	84.3 ± 12.6	81.4 ± 11.1	<0.001
Insulin, μU/mL	6.3 (4.8;8.2]	5.8 (4.5;7.5)	6.7 (5.2;8.7)	<0.001
TC, mg/dL	188.0 ± 33.6	189.3 ± 33.8	187.0 ± 33.4	0.0142
TG, mg/dL	119.0 (92.0;156.0)	128.0 (98.0;171.0)	113.0 (88.0;147.0)	<0.001
HDL-C, mg/dL	46.1 ± 9.9	44.4 ± 9.5	47.2 ± 10.0	<0.001
CRP, mg/dL	0.23 ± 0.60	0.24 ± 0.51	0.22 ± 0.65	0.185
ALT, IU/L	22.1 ± 8.8	25.9 ± 10.0	19.5 ± 6.8	<0.001
AST, IU/L	26.4 ± 6.9	28.1 ± 7.4	25.2 ± 6.2	<0.001
Steroid medication, *n* (%)	5 (0.1)	1 (0.0)	4 (0.1)	1.000
Anticonvulsant medication, *n* (%)	3 (0.0)	2 (0.1)	1 (0.0)	1.000
HTN medication, *n* (%)	351 (6.5)	110 (4.9)	241 (7.6)	0.014
DM medication *n* (%)	13 (0.2)	10 (0.4)	3 (0.1)	1.000
Dyslipidemia medication, *n* (%)	10 (0.2)	7 (0.3)	3 (0.1)	0.790

Data are presented as mean ± standard deviations for continuous variables and number (%) for categorical variables. Abbreviations: WC, waist circumference; BMI, body mass index; TMM, total muscle mass; MBP, mean blood pressure; FPG, Fating plasma glucose; TC, total cholesterol; TG, triglyceride; HDL-C, high-density lipoprotein cholesterol; CRP, C reactive protein; ALT, alanine aminotransferase; AST, aspartate aminotransferase; HTN, hypertension; DM, diabetes mellitus.

**Table 2 nutrients-14-02994-t002:** Hazard ratio and 95% confidence interval for incident non–alcoholic fatty liver diseases according to five body composition indices.

Total	Univariable Model		Multivariable Model 1		Multivariable Model 2		Multivariable Model 3	
	HR (95% CI)	*p*	HR (95% CI)	*p*	HR (95% CI)	*p*	HR (95% CI)	*p*
WC, cm	1.07 (1.07–1.08)	<0.001	1.07 (1.07–1.08)	<0.001	1.07 (1.07–1.08)	<0.001	1.07 (1.06–1.07)	<0.001
Body fat, %	1.05 (1.04–1.05)	<0.001	1.10 (1.09–1.10)	<0.001	1.10 (1.09–1.11)	<0.001	1.09 (1.08–1.09)	<0.001
BMI, kg/m^2^	1.21 (1.19–1.22)	<0.001	1.22 (1.20–1.23)	<0.001	1.22 (1.20–1.24)	<0.001	1.19 (1.18–1.21)	<0.001
TMM/Ht^2^, kg/m^2^	1.21 (1.18–1.24)	<0.001	1.37 (1.33–1.41)	<0.001	1.38 (1.33–1.42)	<0.001	1.32 (1.28–1.36)	<0.001
TMM/BMI	0.56 (0.49–0.64)	<0.001	0.20 (0.16–0.24)	<0.001	0.19 (0.15–0.24)	<0.001	0.26 (0.20–0.33)	<0.001
Men								
WC, cm	1.09 (1.08–1.10)	<0.001	1.09 (1.08–1.10)	<0.001	1.10 (1.09–1.11)	<0.001	1.09 (1.08–1.10)	<0.001
Body fat, %	1.09 (1.08–1.10)	<0.001	1.09 (1.08–1.11)	<0.001	1.10 (1.09–1.11)	<0.001	1.09 (1.07–1.10)	<0.001
BMI, kg/m^2^	1.23 (1.20–1.26)	<0.001	1.23 (1.20–1.26)	<0.001	1.25 (1.22–1.28)	<0.001	1.22 (1.19–1.25)	<0.001
TMM/Ht^2^, kg/m^2^	1.28 (1.27–1.34)	<0.001	1.29 (1.23–1.35)	<0.001	1.30 (1.24–1.36)	<0.001	1.27 (1.21–1.33)	<0.001
TMM/BMI	0.33 (0.25–0.44)	<0.001	0.27 (0.20–0.37)	<0.001	0.24 (0.18–0.34)	<0.001	0.34 (0.25–0.48)	<0.001
Women								
WC, cm	1.07 (1.06–1.07)	<0.001	1.06 (1.06–1.07)	<0.001	1.06 (1.06–1.07)	<0.001	1.06 (1.05–1.07)	<0.001
Body fat, %	1.10 (1.09–1.12)	<0.001	1.10 (1.09–1.11)	<0.001	1.10 (1.09–1.11)	<0.001	1.09 (1.08–1.10)	<0.001
BMI, kg/m^2^	1.21 (1.19–1.23)	<0.001	1.21 (1.19–1.23)	<0.001	1.20 (1.18–1.23)	<0.001	1.18 (1.16–1.21)	<0.001
TMM/Ht^2^, kg/m^2^	1.38 (1.32–1.44)	<0.001	1.417 (1.36–1.48)	<0.001	1.42 (1.36–1.48)	<0.001	1.36 (1.30–1.42)	<0.001
TMM/BMI	0.11 (0.08–0.14)	<0.001	0.13 (0.10–0.18)	<0.001	0.14 (0.10–0.19)	<0.001	0.18 (0.13–0.25)	<0.001

Abbreviations: HR, hazard ratio; CI, confidence interval; WC, waist circumference; BMI, body mass index; TMM, total muscle mass. Model 1: adjusted for age and sex (for total); Model 2: adjusted for variables used in Model 1 plus alcohol drinking status, smoking status, physical activity, and total calorie; Model 3: adjusted for variables used in Model 2 plus mean blood pressure, fasting plasma glucose level, serum total cholesterol, CRP, and ALT levels.

**Table 3 nutrients-14-02994-t003:** Cox regression analysis for incidence of non–alcoholic fatty liver diseases according to the optimal waist circumference values by sex.

		Unadjusted		Model 1		Model 2		Model 3	
		HR (95% CI)	*p*	HR (95% CI)	*p*	HR (95% CI)	*p*	HR (95% CI)	*p*
Total	WC								
	< 80.2	reference		reference		reference		reference	
	≥80.2	2.75 (2.53–2.99)	<0.001	2.76 (2.53–3.01)	<0.001	2.76 (2.53–3.02)	<0.001	2.44 (2.23–2.67)	<0.001
Men	WC								
	<81	reference		reference		reference		reference	
	≥81	2.88 (2.50–3.32)	<0.001	2.87 (2.50–3.31)	<0.001	2.99 (2.58–3.46)	<0.001	2.65 (2.28–3.08)	<0.001
Women	WC								
	<78.5	reference		reference		reference		reference	
	≥78.5	2.93 (2.62–3.28)	<0.001	2.77 (2.47–3.12)	<0.001	2.77 (2.45–3.13)	<0.001	2.54 (2.25–2.87)	<0.001

Abbreviations: WC, waist circumference; HR, hazard ration; CI, confidence interval. Model 1: adjusted for age and sex (for total); Model 2: adjusted for variables used in Model 1 plus alcohol drinking status, smoking status, physical activity, and total calorie; Model 3: adjusted for variables used in Model 2 plus mean blood pressure, fasting plasma glucose level, serum total cholesterol, CRP, and ALT levels.

## Data Availability

The data used in this study are available at the following website; http://www.cdc.go.kr/contents.es?mid=a40504010000 (accessed on 14 January 2022).

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
