# Peer review of "Cutoff Points of Waist Circumference for Predicting Incident Non-Alcoholic Fatty Liver Disease in Middle-Aged and Older Korean Adults"

_nutrients, 2022, doi:10.3390/nu14142994_

Round 1

Reviewer 1 Report

The authors examined the optimal cutoff value of waist circumference (WC) for predicting incident NAFLD in middle-aged and older Korean adults.

WC was the most significant risk factor for predicting incident NAFLD among body composition indicators. The optimal WC cutoff point for predicting incident NALFD was 81cm in men and 78.5cm in women, which might assist in early detection and prevention of NAFLD.

The findings are interesting and important topic. I have several concerns.

1.The authors excluded the subjects that alcohol intake ≥ 30 g per day in men and ≥ 20 g per day in women. How did the authors assess the alcohol intake in the present study?

2.In Table.1, authors showed baseline characteristic of the study population. Authors should add the data of lipid and glucose metabolism such as triglyceride, insulin, and medication related with NAFLD.

3. Authors used height squared-adjusted TMM (TMM/Ht2) and BMI-adjusted TMM (TMM/BMI) for the analysis. What is a significance and rationale of the markers in the present study

4.In Table.3, the authors used three models for adjustment. Why and How did the authors choose the factors?

5.The optimal WC cutoff point for predicting incident NALFD was 81cm in men and 78.5cm in women in Korean adults. It would be better to discuss the difference between gender and races in greater detail.

Reviewer 2 Report

This is a well written paper. It is well known that waist circumference is a predictor of NAFLD,

The paper is important as a cut-off for the Korean population, so my suggestion is that it must be written in the title,

Author Response

Thank you for inviting us to submit a revised draft of our manuscript. We appreciate the time and effort you have dedicated to providing insightful feedback on ways to strengthen our paper. Thus, it is with great pleasure that we resubmit our article for further consideration. We modified the title as follows; “Cutoff Points of Waist Circumference for Predicting Incident Non-Alcoholic Fatty Liver Disease in Korean middle aged and older adults.”

Round 2

Reviewer 1 Report

The revision has improved the manuscript. I have no further concern.